# High Mortality in an Older Japanese Population with Low Forced Vital Capacity and Gender-Dependent Potential Impact of Muscle Strength: Longitudinal Cohort Study

**DOI:** 10.3390/jcm11185264

**Published:** 2022-09-06

**Authors:** Midori Miyatake, Tatsuma Okazaki, Yoshimi Suzukamo, Sanae Matsuyama, Ichiro Tsuji, Shin-Ichi Izumi

**Affiliations:** 1Department of Physical Medicine and Rehabilitation, Tohoku University Graduate School of Medicine, Sendai 980-8575, Japan; 2Center for Dysphagia of Tohoku University Hospital, Sendai 980-8575, Japan; 3Division of Epidemiology, Department of Health Informatics and Public Health, School of Public Health, Tohoku University Graduate School of Medicine, Sendai 980-8575, Japan; 4Department of Physical Medicine and Rehabilitation, Tohoku University Graduate School of Biomedical Engineering, Sendai 980-8575, Japan

**Keywords:** % predicted value forced vital capacity, mortality, muscle strength, older people, cohort study, pneumonia

## Abstract

Generally, weak muscle power is associated with high mortality. We aimed to evaluate the unknown association between % predicted value forced vital capacity (FVC% predicted) and mortality in asymptomatic older people, and the impact of muscle power on this association. We analyzed the Tsurugaya cohort that enrolled Japanese people aged ≥70 for 15 years with Cox proportional hazards model. Exposure variables were FVC% predicted and leg power. The outcome was all-cause mortality. The subjects were divided into quartiles by FVC% predicted or leg power, or into two groups by 80% for FVC% predicted or by the strongest 25% for leg power. Across 985 subjects, 262 died. The males with lower FVC% predicted exhibited higher mortality risks. The hazard ratio (HR) was 2.03 (95% CI 1.30–3.18) at the lowest relative to the highest groups. The addition of leg power reduced the HR to 1.78 (95% CI 1.12–2.80). In females, FVC% predicted under 80% was a risk factor and the HR was 1.67 (95% CI 1.05–2.64) without the effect of leg power. In FVC% predicted <80% males HRs were 2.44 (95% CI 1.48–4.02) in weak and 1.38 (95% CI 0.52–3.64) in strong leg power males, relative to ≥80% and strong leg power males. Low FVC% predicted was associated with high mortality with potential unfavorable effects of weak leg power in males.

## 1. Introduction

Forced vital capacity (FVC) and % predicted value FVC (FVC% predicted) are major indicators of respiratory functions. Several general population studies reported an association between low FVC/FVC% predicted and high mortality [1,2,3,4,5,6]. FVC and FVC% predicted are generally used to diagnose restrictive lung diseases, typically represented by interstitial lung diseases. Patients with interstitial lung diseases are accompanied by high mortality [7,8]. A previous study avoided the effects of interstitial lung diseases on mortality by excluding people with respiratory symptoms and showed an association between low FVC and high mortality in general adults [9]. However, another study that analyzed older people with a rather short follow-up period could not show an association [10]. Presently, we only have limited information about the association between FVC% predicted and long-term mortality in community-dwelling older people without respiratory symptoms.

Many studies reported an association between extremity muscle weakness and high mortality [11,12,13,14,15,16]. Extremity muscle strength and respiratory function are moderately correlated with respiratory muscle strength [17,18,19,20]. Respiratory muscle weakness is associated with the onset of and potentially death by pneumonia in older people [21]. Presently, the effects of muscle strength on the association between FVC% predicted and mortality are unclear.

The main objective of this study was to investigate the association between low FVC% predicted and high mortality in community-dwelling asymptomatic older people. Our secondary objective was to determine whether weak extremity muscle power has an unfavorable effect on the association between low FVC% predicted and high mortality. We hypothesized that low FVC% predicted was associated with high mortality in general asymptomatic older people. We next hypothesized that extremity muscle weakness has an unfavorable effect on the association. To achieve the above objectives, we analyzed a longitudinal cohort, the Tsurugaya project.

## 2. Materials and Methods

### 2.1. Participants

We used the baseline data from 2002 on the comprehensive geriatric assessments conducted for older people aged ≥70 years in the Tsurugaya project [22,23]. We excluded potential interstitial lung disease patients by evaluating dyspnea as a respiratory symptom. Dyspnea was evaluated by breathing through an external circuit with an inspiratory resistive load of 10, 20, and 30 cmH_2_O/L/s and was assessed by the modified Borg scale [24,25]. This is a category scale in which the subject selects a number from 0 (no dyspnea) to 10 (maximum dyspnea). Subjects selected 2 or greater at baseline breathing without inspiratory resistance for 1 minute were excluded.

### 2.2. Measurements

Spirometry (OST 80A, Chest Co., Tokyo, Japan) was measured 3 times, and the best trial was reported. The FVC% predicted values were calculated by age, gender, and height [26]. Initial measurements were performed in 2002, following American Thoracic Society recommendations [27], applying reference values and a cut-off value for FVC as 80% to FVC% predicted, as the Japanese respiratory society published in 2001. Current Japanese reference values were published in 2014; however, we applied the old reference values in this study to keep consistency within the Tsurugaya project [24,26]. In each gender, the subjects were divided into 4 groups based on quartiles of FVC% predicted or 2 groups (<80% and ≥80%) [26]. Leg extension power (w/kg) was measured on a horizontal leg extension apparatus (Combi Anaeropress3500, Tokyo, Japan) [28]. The average of the 2 strongest leg power measurements among 5 trials was divided by the bodyweight [28]. In each gender, the subjects were divided into 4 groups based on quartiles or into 2 groups, the strongest 25% group and other groups. Sociodemographic and medical history data were collected using questionnaires. The questionnaire included age, sex, medical history (pneumonia, bronchial asthma, cancer, myocardial infarction, stroke, diabetes, hypertension), smoking, alcohol consumption, educational levels, and marital status. The smoking status and alcohol consumption were categorical variables; subjects were classified into current, past, or never smoking groups and non-alcohol, current, or past consumption groups. The educational periods categorized the subjects into <18- or ≥18-year groups. Depressive symptoms were evaluated with the Japanese version of the 30-point Geriatric Depression Scale (GDS) [29], and cognitive function was measured with the Japanese version of the Mini-Mental State Examination (MMSE) [30]. The serum albumin and high sensitive C-reactive protein (hs-CRP) concentrations were assessed in blood samples collected under non-fasting conditions by a clinical testing laboratory. Body mass index (BMI, kg/m^2^) was calculated.

### 2.3. Mortality Follow-Up

The outcome of the study was all-cause mortality. Data regarding death or migration were received from the Sendai Municipal Authority. The follow-up period was from 30 March 2003 to 1 July 2018.

### 2.4. Statistical Analysis

The baseline characteristics of 4 groups divided by FVC% predicted were compared by chi-square tests for categorical variables and one-way ANOVA for continuous variables. The cumulative survival rate of the 4 groups was compared according to the FVC% predicted using the Kaplan–Meier method and log-rank tests. The Cox proportional hazard model was used to calculate the hazard ratios (HRs) and 95% confidence intervals (CIs) for all-cause mortality during the follow-up period. The highest group was used as the reference category. To investigate the relationships among the 4 groups for FVC% predicted and all-cause mortality, we ran 2 models. Model 1 was a univariate model. Model 2 was adjusted by all potential covariates; potential covariates for adjustment included age, medical histories (pneumonia, bronchial asthma, cancer, stroke, myocardial infarction, diabetes, hypertension), smoking, alcohol consumption, depressive symptoms, cognitive function, educational level, marital status, leg power (divided into 4 groups), BMI, albumin, and hs-CRP. The 1 and 2 models were also applied to investigate the relationship between the 2 groups for FVC% predicted divided by the clinical cut-off value of ≥80% and all-cause mortality. We also evaluated the effect of muscle strength (divided into 4 groups) on the relationship among the 4 groups for FVC% predicted and all-cause mortality. Model 1 was a univariate model. Leg power was added as an explanatory variable in Model 2. Model 3 was adjusted by all potential covariates. The 1–3 models were also applied to investigate the effect of muscle strength divided by the strongest 25% on the relationship between FVC% predicted divided by ≥80% and all-cause mortality. To evaluate the relationship between muscle strength, FVC% predicted, and mortality, we divided the subjects into 2 groups by ≥80% for FVC% predicted and by the strongest 25% for muscle strength, respectively, a total of 4 groups. Model 1 was a univariate model. Model 2 was adjusted by all potential covariates. We also conducted a sensitivity analysis to test the robustness of the association between FVC% predicted and all-cause mortality. The above statistical analyses were performed using SPSS software, version 24.0 (IBM Corp., Armonk, NY, USA). We evaluated the power of the main result by the post hoc power analysis using Power and Precision 4.1 (Biostat, Englewood, NJ, USA). *p* values less than 0.05 were considered significant.

## 3. Results

We invited all residents who were ≥70 years old to the Tsurugaya project in 2002 (*n* = 2730). Of 1198 participating subjects, 1175 provided written informed consent. Figure 1 shows a study flow chart. Nineteen subjects without spirometry data and 95 subjects with missing or incomplete leg extension power data were excluded. We excluded eight subjects with a Mini-Mental State Examination (MMSE) score below 10 or missing to maintain the reliability of the spirometry data. Nineteen subjects lacked high-sensitive C-reactive protein (hs-CRP) data. We excluded 49 subjects with respiratory symptoms to exclude potential interstitial lung disease patients. Finally, 985 subjects were analyzed.

The subjects were divided by gender and then into four groups by FVC% predicted (Quartile1: FVC% predicted ≥100.5 [male], ≥114.5 [female]; Quartile2: ≥89.2 to <100.5 [male], ≥100.9 to <114.5 [female]; Quartile3: ≥78.3 to <89.2 [male], ≥87.7 to <100.9 [female]; and Quartile4: <78.3 [male], <87.7 [female]). Quartile 1 was the highest and Quartile 4 was the lowest FVC% predicted group. Table 1 shows the characteristics recorded in the baseline survey. Among the 985 participants, the proportion of males was 42.6%, and the mean age (standard deviation [SD]) was 75.6 (4.8) years. The variables that showed significant differences among the four groups were age, education levels, the incidence of past histories of pneumonia and bronchial asthma, leg extension power, and hs-CRP.

During the follow-up period of 13,011 person-years, of the 420 males, 154 (36.7%) died, and of the 565 females, 108 (19.1%) died. Figure 2 shows Kaplan–Meier survival curves according to the FVC% predicted in males (Figure 2A) and females (Figure 2B). The cumulative survival rates were significantly lower in the lowest FVC% predicted group Quartile 4 in males (log-rank test, *p* < 0.005), but there were no differences in females (log-rank test, *p* = 0.193). The number at risk in Figure 2 showed the number of individuals at risk of experiencing an event for each cohort during the follow-up period every 2 years. Figure 2 also showed the number of participants lost to follow-up every 2 years. These data suggest censoring occurred randomly, independent of the event.

Kaplan–Meier curves showing the cumulative survival rates according to Quartiles of % predicted value forced vital capacity (FVC% predicted). The cumulative survival rates were significantly lower in the lowest FVC% predicted group Quartile 4 in males (log-rank test, *p* < 0.005) but not in females (log-rank test, *p* = 0.193). The comparison between Quartile 4 and other groups: Q4 versus Q1; *p* = 0.001, Q4 versus Q2; *p* = 0.040, Q4 versus Q3; *p* = 0.006 in males. Q1; FVC% predicted ≥100.5 (male), ≥114.5 (female). Q2; FVC% predicted ≥89.2 to <100.5 (male), ≥100.9 to <114.5 (female). Q3; FVC% predicted ≥78.3 to <89.2 (male), ≥87.7 to <100.9 (female). Q4; FVC% predicted <78.3 (male), <87.7 (female). The number at risk shows the number of individuals at risk of experiencing an event for each group every 2 years. The lost to follow-up ratio shows the number of participants lost to follow-up for each group every 2 years.

Table 2 shows the association between the FVC% predicted divided into quartiles and mortality in males. Since the Kaplan–Meier survival curve of females divided into four groups by FVC% predicted could not show differences, we showed the data of males in the following analysis using quartile FVC% predicted. The highest FVC% predicted group (Q1) was set as a reference. Model 1 was a univariate model, and Model 2 was adjusted for multiple covariates. In males, lower FVC% predicted was inversely associated with higher mortality; the multivariate-adjusted HRs (95% CIs) was 2.12 (1.31 to 3.43) for the lowest FVC% predicted group Q4, which was significantly associated with higher mortality (*p* for trend = 0.010 in Model 3).

We then conducted sensitivity analyses for males. The results did not change after excluding cases who died in the first 3 years of follow-up (Appendix A).

Males were divided into four groups by leg power: The strongest Quartile 1 (w/kg): ≥15.8; Quartile 2: ≥13.1 to <15.8; Quartile 3: ≥10.8 to <13.1 and the weakest Quartile 4: <10.8. Table 3 shows the effect of muscle strength on the association between FVC% predicted and mortality. Model 1 showed that the HR (95% Cl) of the lowest FVC% predicted group Q4 was 2.03 (1.30 to 3.18). In Model 2, the leg extension power was added to Model 1 as an explanatory variable, and the strongest leg power group Q1 was set as a reference. The HR of the lowest FVC% predicted group Q4 reduced to 1.78 (1.12 to 2.80) in Model 2 from 2.03 in Model 1. Leg power was significantly associated with mortality independently from FVC% predicted (*p* for trend = 0.025 in model 2). The weakest leg power group Q4 showed a HR of 1.67 (1.02 to 2.74), significantly associated with higher mortality.

To evaluate the relationship between the FVC% predicted, leg power, and mortality, we divided the subjects into a total of four groups by the clinical cut-off value of FVC% predicted (<80% group and ≥80% group) and by leg power (the strongest 25% group and the other groups), respectively (Table 4). The group with high FVC% predicted and strong leg power was set as a reference. Model 1 was a univariate model, and Model 2 was adjusted for multiple covariates. In Model 2, among low FVC% predicted males, the HRs were 2.69 (1.56 to 4.66) in the weak and 1.55 (0.58 to 4.15) in the strong leg power groups. The HRs showed similar trends in Models 1 and 2. The above findings suggest the potential unfavorable effects of weak leg power on the mortality of males among the low FVC% predicted subjects.

We then used the FVC% predicted 80% and divided the subjects into two groups. Appendix A shows the association between the FVC% predicted (<80% group and ≥80% group) and mortality in males and females. The higher FVC% predicted ≥80% group was set as a reference. In both males and females, lower FVC% predicted was significantly associated with higher mortality. Males: HR 1.58 (1.11 to 2.24) and females: 1.87 (1.11 to 3.14) in Model 2.

Finally, the subjects were divided by gender and then into two groups by leg power (the strongest 25% group and the other groups). Appendix A shows the effect of muscle strength on the association between FVC% predicted (<80% group and ≥80% group) and mortality. In females, Model 1 showed that the HR (95% Cl) of the low FVC% predicted group was 1.67 (1.05 to 2.64). In Model 2, the leg extension power was added to Model 1 as an explanatory variable, and the strong leg power group was set as a reference. The HRs of the low FVC% predicted groups essentially did not change, 1.67 (1.05 to 2.67) in Model 2 and 1.81 in (1.08 to 3.02) Model 3. Leg power was not associated with mortality. The weak leg power group showed a HR of 0.98 (0.63 to 1.52) in Model 2 and 1.05 (0.61 to 1.79) in Model 3. In males, the effect of muscle strength on the association between FVC% predicted and mortality showed a similar trend to Table 3, which was divided into four groups. Overall, the above findings suggest that weak leg power did not affect the mortality of females among the low FVC% predicted group.

## 4. Discussion

Lower FVC% predicted was associated with a higher risk of all-cause mortality in community-dwelling older males without respiratory symptoms. FVC% predicted under 80% was a risk factor for all-cause mortality in females. Weak leg power may have the potential to unfavorably affect the association in males with low FVC% predicted but not in females.

This study showed that weak leg power was associated with high mortality in community-dwelling older males. Previous studies showed an association between weak extremity muscle power, including leg power and high mortality [11,12,13,14,15,16]. We could not find such an association in females. However, a previous study could not show an association in older Japanese females [13]. This may reflect the differences in ethnicity or lifestyles between Japanese/east Asians and people in other countries.

Previous studies showing an association between low FVC and high mortality did not mention factors that can affect the risk [1,2,3,4,5,6,9,10]. This study showed a dose-response relationship between higher FVC% predicted and lower male mortality. Furthermore, when leg power was put into the model as an explanatory variable, the variable was independently associated with mortality and reduced the HR of the low FVC% predicted. This suggests that weak leg power may predict the high mortality of low FVC% predicted. Indeed, analysis of the subjects divided into four groups by the FVC% predicted (two groups) and leg power (two groups) showed a trend of high mortality risk in the weak leg power group than in the strong leg power group among low FVC% predicted males. However, the number of males with low FVC% predicted and strong leg power was insufficient.

The cause of death from low FVC in people without respiratory symptoms is unclear [9]. For extremity muscles, weak leg power was associated with high-pneumonia mortality in older males, but the mechanisms were unclear [31]. Respiratory function and extremity muscle power moderately correlate with respiratory muscle strength [17,18,19,20]. The respiratory muscle strength has a clear relationship with pneumonia. Strong respiratory muscles generate effective coughing, clear the airways, and prevent pneumonia [18,21,32,33]. Indeed, respiratory muscle weakness was a risk factor for the onset of and possibly death by pneumonia in older people [21]. Muscle weakness is related to inflammation, including pneumonia [34,35,36,37]. Thus, the association between weak extremity muscle power and pneumonia might be mediated through respiratory muscle weakness. Likewise, low FVC% predicted might be associated with pneumonia through respiratory muscle weakness. From the point of view of muscle weakness, asymptomatic neuromuscular diseases can cause death. Furthermore, considering respiratory muscle weakness, diaphragmatic dysfunction, such as diaphragmatic nerve paralysis, can also cause death.

Females showed an association between low FVC% predicted under 80% and high mortality, and leg power did not affect this association. Thus, in older Japanese females, associations between respiratory muscle strength, leg power, and lung function might be weak, or other unknown factors might be involved.

A previous study suggested that FVC was not associated with mortality in older people without respiratory symptoms [10]. This discrepancy is possibly due to their 8-year follow-up. In contrast, it was 15 years in our study, in which the analysis was of FVC instead of FVC% predicted, or did not include dyspnea as a respiratory symptom.

Forced expiratory volume in 1 second (FEV1.0) is another major indicator of respiratory functions with its well-known association with mortality, and its value is greatly worsened by smoking [1,5]. Since the effects of smoking on FEV1.0 were extremely great [1,3,5], we had difficulties analyzing the effects of muscle power on the association between FEV1.0 and mortality.

We calculated the sample size of our main result; the lower FVC% predicted males exhibited higher mortality risks. The power for males’ Q1 and Q4 hazard ratios was 0.89, which suggests a sufficient sample size for the analysis.

This study has some limitations. First, this study could not examine the association between FVC% predicted and cause-specific mortality. Therefore, other data sets will be required to identify the association. Second, the ethnicity was limited to Japanese. Third, we could not identify factors that affect the high mortality in females with low FVC% predicted. Fourth, we could not completely exclude potential interstitial lung disease patients since they can present asymptomatic and with normal pulmonary function tests.

The present study suggests that low FVC% predicted is associated with an increased risk of all-cause mortality in community-dwelling older people without respiratory symptoms. In addition, this study suggests that weak leg power predicts a high risk of death due to the low FVC% predicted in older males. Hence, we may highlight the increased risk to such males in various situations, such as in clinical settings and medical examinations.

## Figures and Tables

**Figure 1 jcm-11-05264-f001:**
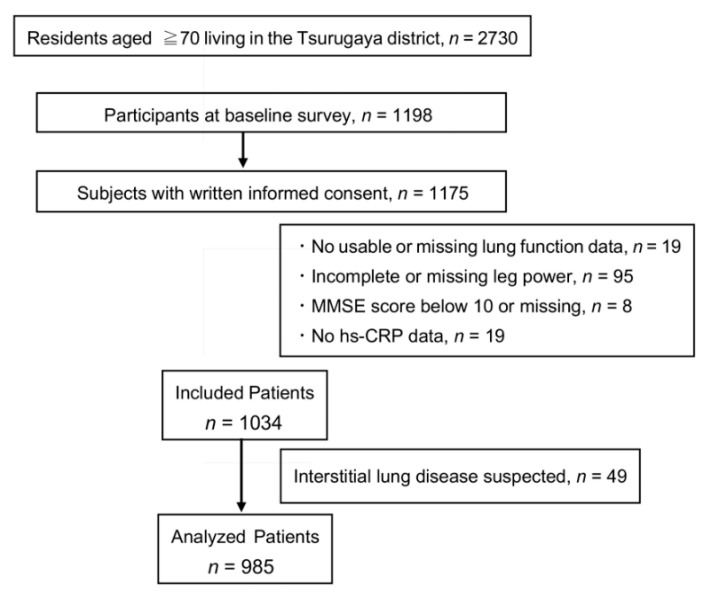
Inclusion and exclusion of the study participants. MMSE = Mini-Mental State Examination, hs-CRP = high-sensitive C-reactive protein.

**Figure 2 jcm-11-05264-f002:**
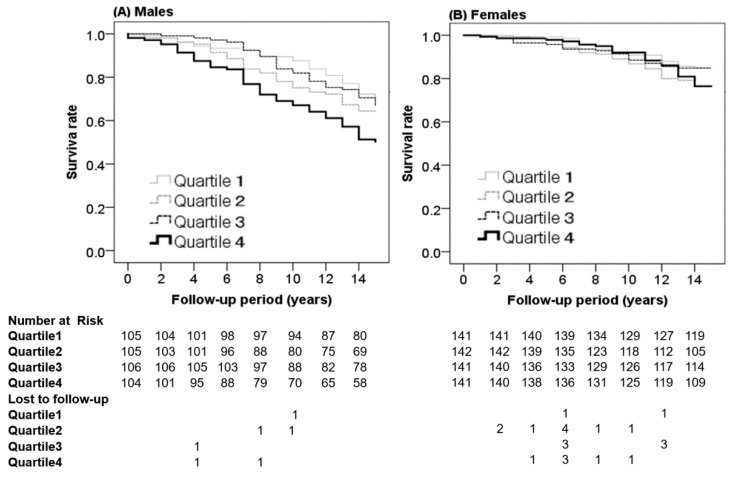
Kaplan–Meier survival curves showing the cumulative survival rate according to the FVC% predicted in males (**A**) and females (**B**).

**Table 1 jcm-11-05264-t001:** Baseline characteristics of the study participants according to FVC% predicted (*n* = 985).

Characteristics		FVC% Predicted				
	Overall	Q1 ^‡^	Q2 ^§^	Q3 ^‖^	Q4 **	*p* *
Number of participants	985	246	247	247	245	
Age, mean (SD)	75.6 (4.8)	75.9 (5.2)	75.4(4.7)	75.1 (4.2)	76.1 (4.8)	0.047
Men, *n* (%)	420 (42.6)	105 (42.7)	105 (42.5)	106 (42.9)	104 (42.4)	1.000
Medical history, *n* (%)						
Pneumonia	92 (9.3)	17 (6.9)	23 (9.3)	16 (6.5)	36 (14.7)	0.006
Bronchial asthma	56 (6.0)	8 (3.3)	10 (4.0)	14 (5.7)	27 (11.0)	0.001
Cancer	65 (6.6)	18 (7.3)	15 (6.1)	18 (7.3)	14 (5.7)	0.845
Myocardial infarction	108 (11.0)	24 (9.8)	29 (11.7)	29 (11.7)	26 (10.6)	0.872
Stroke	46 (4.7)	15 (6.1)	9 (3.6)	14 (5.7)	8 (3.3)	0.341
Diabetes mellitus	138 (14.0)	39(15.9)	33 (13.4)	29 (11.7)	37 (15.1)	0.556
Hypertension	370 (37.6)	77 (31.3)	94 (38.1)	98 (39.7)	101 (41.2)	0.111
Current smoking, *n* (%)	130 (13.2)	33 (13.7)	32 (13.2)	30 (12.4)	35 (14.7)	0.924
Alcohol consumption, *n* (%)	400 (40.6)	107 (45.5)	100 (42.4)	109 (46.6)	84 (36.4)	0.174
Duration of education ≥18 years, *n* (%) ^†^	561 (57.0)	143 (58.4)	145 (58.9)	153 (62.2)	120 (49.6)	0.034
Marital status, *n* (%)	609 (61.8)	156 (63.4)	157 (63.6)	147 (59.5)	149 (61.6)	0.950
Cognitive impairment, mean (SD)	27.4 (2.6)	27.5 (2.5)	27.6 (2.3)	27.2 (2.8)	27.1 (2.8)	0.067
Depressive symptoms, mean (SD)	8.9 (5.4)	8.6 (5.4)	9.0 (5.6)	8.7 (5.6)	9.2 (5.0)	0.553
BMI (kg/m^2^), mean (SD)	23.8 (3.3)	23.5 (3.0)	23.8 (3.1)	24.1 (3.2)	23.7 (3.8)	0.135
Leg extension power (w/kg), mean (SD)	10.1 (4.5)	10.8 (4.8)	10.4 (4.4)	9.9 (4.5)	9.0 (4.1)	<0.001
Albumin, (g/dL), mean (SD)	4.3 (0.3)	4.3 (0.3)	4.3 (0.3)	4.3 (0.3)	4.3 (0.3)	0.634
hs-CRP (ng/mL), mean (SD)	1790 (5325)	965 (1581)	2068 (7276)	1563 (4459)	2565 (6084)	<0.001

FVC% predicted = % predicted value forced vital capacity; SD = standard deviation; BMI = body mass index; hs-CRP = high sensitive C-reactive protein; ANOVA = analysis of variance. * Obtained by using chi-squared test for variables of proportion and one-factor ANOVA for continuous variables. ^†^ Age at last school graduation 18 years. ^‡^ Q1; FVC% predicted ≥100.5 (male), ≥114.5 (female). ^§^ Q2; FVC% predicted ≥89.2 to <100.5 (male), ≥100.9 to <114.5 (female). ^‖^ Q3; FVC% predicted ≥78.3 to <89.2 (male), ≥87.7 to <100.9 (female). ** Q4; FVC% predicted <78.3 (male), <87.7 (female).

**Table 2 jcm-11-05264-t002:** Association between FVC% predicted divided into quartiles and mortality *.

	Males (420)				
	Q1 ^§^	Q2 **	Q3 ^††^	Q4 ^‡‡^	*p*-trend
**Number of cases**	105	105	106	104	
**Number of death**	31	37	35	51	
**Model 1 ^†^**	1.0 (ref.)	1.31 (0.81–2.11)	1.14 (0.70–1.85)	2.03 (1.30–3.18)	0.006
**Model 2 ^‡^**	1.0 (ref.)	1.48 (0.90–2.43)	1.18 (0.70–1.98)	2.12 (1.31–3.43)	0.010

FVC% predicted = % predicted value forced vital capacity; BMI = body mass index; hs-CRP = high sensitive C-reactive protein, ref = reference. * Hazard ratio (95% confidence interval). ^†^ Model 1: univariate model. ^‡^ Model 2: adjusted for age, medical history (pneumonia, bronchial asthma, cancer, stroke, myocardial infarction, diabetes, hypertension), smoking, alcohol consumption, depressive symptoms, cognitive function, educational level, marital status, leg extension power, BMI, albumin and hs-CRP. ^§^ Q1; FVC% predicted ≥100.5. ** Q2; FVC% predicted ≥89.2 to <100.5. ^††^ Q3; FVC% predicted ≥78.3 to <89.2. ^‡‡^ Q4; FVC% predicted <78.3.

**Table 3 jcm-11-05264-t003:** The effect of muscle strength on the association between FVC% predicted divided into quartiles and mortality *.

	Males (420)				
	Q1 ^‖^	Q2 **	Q3 ^††^	Q4 ^‡‡^	*p*-trend
**Number of cases**					
FVC% predicted	105	105	106	104	
Leg power	107	105	107	101	
**Model 1 ^†^**					
FVC% predicted	1.0 (ref.)	1.31 (0.81–2.11)	1.14 (0.70–1.85)	2.03 (1.30–3.18)	0.006
Leg power		-	-	-	-
**Model 2 ^‡^**					
FVC% predicted	1.0 (ref.)	1.26 (0.78–2.03)	1.06 (0.65–1.72)	1.78 (1.12–2.80)	0.036
Leg power		1.43 (0.87–2.35)	1.79 (1.10–2.90)	1.67 (1.02–2.74)	0.025
**Model 3 ^§^**					
FVC% predicted	1.0 (ref.)	1.48 (0.90–2.43)	1.18 (0.70–1.98)	2.12 (1.31–3.43)	0.010
Leg power		1.41 (0.84–2.38)	1.92 (1.15–3.23)	1.69 (0.96–2.98)	0.039

FVC% predicted = % predicted value forced vital capacity; BMI = body mass index; hs-CRP = high sensitive C-reactive protein, ref = reference. * Hazard ratio (95% confidence interval). ^†^ Model 1: univariate model. ^‡^ Model 2: Model 1+ leg extension power divided into quartiles. ^§^ Model 3: Model 2 adjusted for age, medical history (pneumonia, bronchial asthma, cancer, stroke, myocardial infarction, diabetes, hypertension), smoking, alcohol consumption, depressive symptoms, cognitive function, educational level, marital status, BMI, albumin and hs-CRP. **^‖^** Q1; FVC% predicted ≥100.5 & leg extension power ≥15.8, reference. ** Q2; FVC% predicted ≥89.2 to <100.5 & leg extension power ≥13.1 to <15.8. ^††^ Q3; FVC% predicted ≥78.3 to <89.2 & leg extension power ≥10.8 to <13.1. ^‡‡^ Q4; FVC% predicted <78.3 & leg extension power <10.8.

**Table 4 jcm-11-05264-t004:** The relationship between the leg power (2 groups), FVC% predicted (2 groups), and mortality *.

	Males (420)			
	FVC% predicted ≥80% strong leg power ^§^	FVC% predicted ≥80% weak leg power **	FVC% predicted <80% strong leg power	FVC% predicted <80% weak leg power
**Number of cases**	91	205	16	108
**Number of death**	22	75	5	52
**Model 1 ^†^**	1.0 (ref.)	1.63 (1.02–2.63)	1.38 (0.52–3.64)	2.44 (1.48–4.02)
**Model 2 ^‡^**	1.0 (ref.)	1.69 (1.01–2.82)	1.55 (0.58–4.15)	2.69 (1.56–4.66)

FVC% predicted = % predicted value forced vital capacity; BMI = body mass index; hs-CRP = high sensitive C-reactive protein, ref = reference. * Hazard ratio (95% confidence interval). ^†^ Model 1: univariate model. ^‡^ Model 2: adjusted for age, medical history (pneumonia, bronchial asthma, cancer, stroke, myocardial infarction, diabetes, hypertension), smoking, alcohol consumption, depressive symptoms, cognitive function, educational level, marital status, BMI, albumin and hs-CRP. ^§^ Leg extension power ≥15.8 w/kg. ** Leg extension power <15.8 w/kg.

## Data Availability

Data are available upon reasonable request.

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
