# Peer review of "High Mortality in an Older Japanese Population with Low Forced Vital Capacity and Gender-Dependent Potential Impact of Muscle Strength: Longitudinal Cohort Study"

_jcm, 2022, doi:10.3390/jcm11185264_

Round 1
Reviewer 1 Report
I have read with interest the paper. The objective was to evaluate the association between FVC and mortality and the relationship between muscle strength, FVC% predicted, and mortality.
The study is novel and interesting but can be improved.
Minor comments:
- In the title add that the study was a cohort study." High mortality in an older Japanese population with low forced vital capacity and gender-dependent potential impact of muscle strength. Retrospective cohort study".
- The absence of symptoms and normal pulmonary function tests do not exclude the possibility of ILD. Because they can present asymptomatic and with normal pulmonary function tests. In limitations, it should be added that the presence of ILD cannot be completely excluded.
- Which formula did you use to calculate the Spirometry Reference Values? For example Global Lung Function Initiative (GLI)-2012 equations.
- Figure one can be improved. I suggest adding included patients (n= 1034) and analyzed patients (n=985).
- I suggest adding the sample size calculation, to confirm the power of the study.
Major Comments:
- I suggest adding below the Kaplan-Meier curve the number of individuals at risk of experiencing an event for each cohort for specific points in time.
- The number of participants who were lost to follow-up or censored should be described. To evaluate if censoring occurs randomly independent of the event??
- Observing the K-M curve for women, it is estimated that Hazard's assumption of proportionality is not accomplished. The risk of death among females was not constant over time in the quartiles. In this case, should not be used proportional hazards regression. Perhaps you should use some analysis strategy that considers that the assumption of proportionality is not fulfilled
- Tables 2, 3, and 4 are difficult to read or interpret. Perhaps they could be simplified only by analyzing models 3 or 4. To facilitate understanding and reading for readers.
- With the results obtained, I believe that it cannot be affirmed that strong leg power may predict the reduced mortality of low FVC% predicted.
- In my opinion, with the results obtained, we can say that the decrease in muscle strength was associated with higher mortality. But not that increased strength was associated with lower mortality. In the case that you believe that this is not correct, please explain.
Author Response
Response to the reviewer’s comment:
Reviewer 1
Minor comments:
- In the title add that the study was a cohort study." High mortality in an older Japanese population with low forced vital capacity and gender-dependent potential impact of muscle strength. Retrospective cohort study".
Response to the reviewer’s comment:
We added cohort study to the title. Since the Tsurugaya project is a longitudinal cohort, we wrote longitudinal cohort study.
- The absence of symptoms and normal pulmonary function tests do not exclude the possibility of ILD. Because they can present asymptomatic and with normal pulmonary function tests. In limitations, it should be added that the presence of ILD cannot be completely excluded.
Response to the reviewer’s comment:
Thank you for your careful comment. We added the sentences in the Discussion as limitations (page 9, lines 312-314).
- Which formula did you use to calculate the Spirometry Reference Values? For example Global Lung Function Initiative (GLI)-2012 equations.
Response to the reviewer’s comment:
We performed the Initial Spirometry measurements in 2002. We used a formula using reference values, and a cut-off value for FVC as 80% to FVC% predicted as the Japanese respiratory society published in 2001. Current Japanese reference values were published in 2014. This was for birth cohort effects and improved technology. According to reference 26, GLI-2012 equations did not include reference equations for Japanese. Thus, we suppose applying the reference values for Japanese published in 2001 to measurements in 2002 in this study shows the most accurate results. We referred to the reference values in the materials and methods (page2, lines 71-74).
- Figure one can be improved. I suggest adding included patients (n= 1034) and analyzed patients (n=985).
Response to the reviewer’s comment:
Thank you for your suggestion. We added, include, and analyzed patients in Figure 1. This addition improved the clarity of the figure, thank you.
- I suggest adding the sample size calculation, to confirm the power of the study.
Response to the reviewer’s comment:
Thank you for your suggestion. We calculated the sample size of our main result; the lower FVC% predicted males exhibited higher mortality risks. We conducted the post hoc power analysis by the Power and precision 4.1 (Biostat, Englewood, NJ, USA). The power for males' Q1 and Q4 hazard ratios was 0.89, which suggests a sufficient sample size for the analysis. We wrote these issues in the Materials and Methods (page 3, lines124-126) and the discussion (page 8, lines 305-307).
Major Comments:
- I suggest adding below the Kaplan-Meier curve the number of individuals at risk of experiencing an event for each cohort for specific points in time.
Response to the reviewer’s comment:
We added the number of individuals at risk of experiencing an event every 2-year for each cohort in Figure 2. They are described as Number at Risk in Figure 2.
- The number of participants who were lost to follow-up or censored should be described. To evaluate if censoring occurs randomly independent of the event??
Response to the reviewer’s comment:
Thank you for your in-depth comment. We described the number of participants lost to follow-up every 2 years in Figure 2. Together with a response to previous major comment 1, these data suggest censoring occurred randomly independent of the event. We described these issues in the results (page 4, lines 165-166).
- Observing the K-M curve for women, it is estimated that Hazard's assumption of proportionality is not accomplished. The risk of death among females was not constant over time in the quartiles. In this case, should not be used proportional hazards regression. Perhaps you should use some analysis strategy that considers that the assumption of proportionality is not fulfilled
Response to the reviewer’s comment:
Thank you for pointing out this critical issue. We agree with your comment. Since Hazard's assumption of proportionality was accomplished for males but not for females, we assumed it is not appropriate to show the data together in the following sections. In addition, to facilitate readers' interpretation, we simplified tables to show the quartile data of males only. We believe this simplification made this study easier to understand and strengthened our main findings.
However, another finding in this study was that FVC% predicted under 80% was a risk factor for females, which was not unfavorably affected by muscle weakness. Therefore, we added the leg power divided into 2 groups (the strongest 25% group and the other groups) as an explanatory variable on the association between FVC% predicted (<80% group and ≥80% group) and mortality. This analysis suggested that weak leg power did not affect an association between FVC% predicted and mortality in females. We showed the data as supplemental table 3 (page 7, lines 237-249).
- Tables 2, 3, and 4 are difficult to read or interpret. Perhaps they could be simplified only by analyzing models 3 or 4. To facilitate understanding and reading for readers.
Response to the reviewer’s comment:
We are very sorry for the complicated tables. As mentioned above, we simplified the tables 2, 3, and 4 for males only. Also, we excluded the model adjusted by age. We believe these simplifications facilitate understanding and reading for readers, thank you for your creative comment.
- With the results obtained, I believe that it cannot be affirmed that strong leg power may predict the reduced mortality of low FVC% predicted.
Response to the reviewer’s comment:
You are right. Please read our response to your following comment.
- In my opinion, with the results obtained, we can say that the decrease in muscle strength was associated with higher mortality. But not that increased strength was associated with lower mortality. In the case that you believe that this is not correct, please explain.
Response to the reviewer’s comment:
Thank you for your careful and in-depth interpretation of our data. We agree with your opinion. We changed our description as weak muscle power was associated with higher mortality. This time, our objective was to investigate the association between low FVC% predicted and high mortality, and the effects of muscle power on this association. Therefore, we limited the description of weak muscle power to low FVC% predicted subjects only. We changed our description throughout the manuscript from abstract, introduction, results, to discussion.
Reviewer 2 Report
In this study, the authors set out to find the association between the low forced vital capacity (FVC) predicted and high mortality in the community of older people with no symptoms. The authors also plan to test whether muscle power affects this association. It would be helpful if the authors could state a hypothesis to be tested. The author’s found the association of that lower FVC% is linked to a high risk of mortality. Also, weak leg power was associated with high mortality in older males but not females.
Overall, the data appear to be diligently obtained, are transparently described, and are an important contribution to the field. Authors could expand the discussion into more general considerations about respiratory diseases associated with the FVC and muscle strength.
Author Response
Response to the reviewer’s comment:
Reviewer 2
- In this study, the authors set out to find the association between the low forced vital capacity (FVC) predicted and high mortality in the community of older people with no symptoms. The authors also plan to test whether muscle power affects this association. It would be helpful if the authors could state a hypothesis to be tested. The author’s found the association of that lower FVC% is linked to a high risk of mortality. Also, weak leg power was associated with high mortality in older males but not females.
Response to the reviewer’s comment:
Thank you for your suggestion. We stated our hypothesis in the introduction (page 2, lines 53-55). This addition greatly helps readers to understand the data, thanks again.
- Overall, the data appear to be diligently obtained, are transparently described, and are an important contribution to the field. Authors could expand the discussion into more general considerations about respiratory diseases associated with the FVC and muscle strength.
Response to the reviewer’s comment:
Thank you for your in-depth comment. We discussed our general consideration about diseases associated with the FVC and muscle strength in the discussion (page 8, lines 289-291).
Round 2
Reviewer 1 Report
The authors have responded to each of the comments correctly. The modifications made have improved the quality of the manuscript and made it easier for readers to understand.